

# The Spring Transition of the North Pacific Jet and its Relation to Deep Stratosphere-to-Troposphere Mass Transport over Western North America

Melissa Leah Breeden[1], Amy Hawes Butler[1], John Robert Albers[2], Michael Sprenger[3], Andrew O'Neil Langford[1]

[1]Chemical Sciences Laboratory, NOAA Earth System Research Laboratories, Boulder, 80305, United States of America
[2]Physical Sciences Laboratory, NOAA Earth System Research Laboratories, Boulder, 80305, United States of America
[3]Institute for Atmosphere and Climate Science, ETH Zürich, Zurich, Switzerland

*Correspondence to*: melissa.breeden@noaa.gov

**Abstract.** Stratosphere-to-troposphere mass transport to the planetary boundary layer (STT-PBL) peaks over the western United States during boreal spring, when deep stratospheric intrusions are most frequent. The tropopause-level jet structure modulates the frequency and character of intrusions, although the precise relationship between STT-PBL and jet variability has not been extensively investigated. In this study, we demonstrate how the north Pacific jet transition from winter to summer leads to the observed peak in STT-PBL. We show that the transition enhances STT-PBL through an increase in storm track activity which produces highly-amplified Rossby waves and more frequent deep stratospheric intrusions over western North America. This dynamic transition coincides with the gradually deepening planetary boundary layer, further facilitating STT-PBL in spring. We find that La Niña conditions in late winter are associated with an earlier jet transition and enhanced STT-PBL due to deeper and more frequent tropopause folds. An opposite response is found during El Niño conditions. ENSO conditions also influence STT-PBL in late spring/early summer, during which time La Niña conditions are associated with larger and more frequent tropopause folds than both El Niño and ENSO neutral conditions. These results suggest that knowledge of ENSO state and the north Pacific jet structure in late winter could be leveraged for predicting the strength of STT-PBL in the following months.

## 1 Introduction

The annual cycle of the north Pacific jet drives changes in the circulation response to external perturbations, thereby modifying sensible weather over North America (Fleming et al. 1987; Nakamura 1992; Newman and Sardeshmukh 1998; Lareau and Horel 2012). The jet is strongest during boreal winter due to the overlapping of a polar jet forming via low-level baroclinicity and a subtropical jet forming via outflow from tropical convection (Eichelberger and Hartmann 2007; Jaffe et al. 2011; Li and Wettstein 2012; Christenson et al. 2017). The jet weakens during spring to its summertime minimum as both baroclinicity and tropical convection weaken. This study will focus on how the wintertime jet evolves during spring, which we will refer to as the spring transition. Newman and Sardeshmukh (1998) found that the north Pacific jet transitions from



one contiguous jet core to a double-jet structure in mid-March, including a subtropical branch that extends southeastward from a point near the date line to the southern United States, and a midlatitude branch in the west and central Pacific. This transition coincides with a peak in storm track activity (Nakamura 1992; Hoskins and Hodges 2019) as the background zonal wind and stationary eddy characteristics are strongly linked (Nakamura 1992; Hoskins and Ambrizzi 1993).

How the spring transition affects stratospheric intrusions associated with both potent cyclogenesis and, the focus of this study, stratosphere-to-troposphere mass transport to the planetary boundary layer (STT-PBL), has not been extensively investigated. Skerlak et al. (2014) highlighted a climatological maximum in STT-PBL during boreal spring over western North America, which they attributed to, in part, a deep arid boundary layer while noting a substantial amount of forcing for descent must also be present. The timing in the peak of deep transport differs from peak transport across the tropopause, which is strongest in boreal winter (Sprenger and Wernli 2003; Skerlak et al. 2014), highlighting the unique nature of deep transport events. Given the peak in north Pacific storm track activity during boreal spring and corresponding peak in STT-PBL, we hypothesize that the invigoration of the storm track from late winter to spring produces stronger stratospheric intrusions, enhancing STT-PBL.

Natural, non-local sources of ozone to the surface need to be understood and accounted for when creating exceedance limits above the background level for the National Ambient Air Quality Standard (NAAQS). Consistent with the typical seasonal evolution of STT-PBL presented by Skerlak et al. 2014, cases of deep stratospheric ozone intrusions over the western United States have focused predominantly on boreal spring, when intrusions can contribute substantially to the tropospheric ozone budget (Staley 1962; Langford et al. 2009; Langford et al. 2012; Lefohn et al. 2012; Lin et al 2012; Lin et al. 2015; Knowland et al. 2017; Langford et al. 2017; Skerlak et al. 2019). Deep intrusions are commonly observed on the southwest edge of cyclonic potential vorticity (PV) anomalies associated with deep mid-tropospheric troughs, where air is descending along isentropic surfaces, transporting filaments of PV- and ozone-rich stratospheric air into the troposphere (Reed and Danielson 1959; Danielson 1964; Danielson 1968; Shapiro 1980; Keyser and Shapiro 1986; Sprenger et al. 2007; Gettelman et al. 2011). From these case studies, it is evident that several synoptic situations, with the common element of highly-amplified flow and often Rossby wave breaking, can facilitate STT-PBL (e.g., Sprenger et al. 2007). However, there is a clear peak in transport during boreal spring, suggesting a unique set of conditions exist during the spring transition that are conducive to deep transport.

Some studies have noted spring seasons with exceptionally elevated STT-PBL (Lin et al. 2015; Knowland et al. 2017) while some are characterized by relatively minimal STT-PBL (Lin et al. 2015). We will demonstrate that this interannual variability of STT-PBL during spring is related to the timing of the spring transition and, often, the state of the El Niño Southern Oscillation (ENSO). ENSO phase can influence the state of the north Pacific jet (Renwick and Wallace 1996; Shapiro et al. 2001; Martius et al., 2007; Breeden et al. 2020) and STT, although cross-tropopause transport and deep transport display inconsistent responses to ENSO phase (Langford et al. 1998; Zeng and Pyle 2005; Voulgarakis et al. 2011; Lin et al. 2014;



Neu et al. 2014; Lin et al. 2015; Albers et al. 2018). In the mid-troposphere, El Niño conditions can enhance STT in much of the free troposphere through shallow folds along the stronger, extended jet. Conversely, La Niña conditions have been associated with enhanced stratospheric contributions to surface ozone at several monitoring stations located in the western United States, suggesting there is stronger STT-PBL during La Niña conditions compared to El Niño (Lin et al. 2015).


While preliminary results suggest a relationship between ENSO, north Pacific jet variability and STT-PBL over the western US, further investigation of the linkages between these factors is warranted. For instance, changes in specific characteristics of tropopause folds by ENSO phase – such as their vertical and lateral extent, and their frequency – have not been considered, nor have these changes been explicitly linked to variations in STT-PBL. It is the objective of this study to address these sources

of STT-PBL variability using feature-based products designed to study deep transport (Sprenger et al. 2017). We focus on how ENSO modifies the seasonal transition of the north Pacific jet, thereby affecting the timing, frequency and characteristics of stratospheric intrusions and STT-PBL. Section 2 presents the data and methods used for analysis, Section 3.1 presents characteristics of the north Pacific jet transition, Section 3.2 relates the transition to STT-PBL, and Section 3.3 explores the influence of ENSO on the seasonal transition and STT-PBL. Section 4 offers a discussion of results and concluding remarks.

**2. Data and Methods**

**2.1 Data**

Zonal and meridional wind on pressure levels are calculated using the Japanese Reanalysis-55 dataset (Kobayashi et al. 2015), at 2.5 X 2.5° horizontal resolution at 6-hour time intervals, from February – June 1958-2017, and the European Centre for Medium-Range Forecasting (ECMWF) ERA-Interim dataset (Dee et al. 2011) at 1 X 1° horizontal resolution at 6-

hour time intervals from February – June 1979-2017. The 60-year JRA-55 reanalysis is used to compare the spring transition for as many ENSO events as possible, while ERA-Interim is used to be consistent with the transport and tropopause fold diagnostics that were derived from ERA-Interim reanalysis. The state of ENSO was evaluated using the monthly Oceanic Niño Index (ONI) which is based on a threshold of +/- 0.5ºC (positive indicating El Niño, negative indicating La Niña) averaged over the Niño 3.4 region (5ºN-5ºS, 120º-170ºW; NOAA CPC).


STT-PBL, calculated and presented in Skerlak et al. (2014), was accessed at 1 X 1° horizontal resolution and 6-hourly time intervals from February - June 1980-2016. STT-PBL occurs when trajectories that originate in the stratosphere within the four days prior to the crossing of the tropopause reach the pressure level below the PBL. PBL height was determined using the six-hour forecast by the ECMWF model. Each trajectory that originates in the stratosphere and crosses the boundary layer

represents a fixed amount of mass transport: $\Delta m \approx \frac{1}{g}(\Delta x)^2 \Delta p \approx 6.52 \times 10^{11} kg$ where $\Delta x = 80\ km$ and $\Delta p = 30\ hPa$ in the extratropics (Skerlak et al. 2014).



For a measure of the depth of stratospheric intrusions that lead to STT-PBL over western North America, the tropopause fold identification scheme developed by Skerlak et al. (2015) was used at 6-hourly temporal resolution and 1 X 1° horizontal

resolution. Building upon the fold identification scheme introduced by Sprenger et al. (2003), tropopause folds were identified by first distinguishing air of stratospheric and tropospheric origin on a potential vorticity basis using a three-dimensional labelling scheme. Folds of stratospheric origin are then identified when there were multiple crossings of the tropopause (defined using surfaces of the +/- 2-PVU surface and 380 K potential temperature, tropopause marked at whichever surface is lower), observed within a single vertical profile. At locations where folds were identified, the minimum and maximum pressure

the fold reached was recorded (Fig. S1). To examine how the jet transition, and later ENSO phase, affects the depth of tropopause folds, we tracked the maximum pressure a fold, if identified over the western US (box in Fig. S1), reached for each 6-hour timestep. The size of folds deeper than 400 hPa was also tracked by counting the number of grid points over the region that were characterized by a maximum pressure greater than 400 hPa. The number of grid points meeting this criterion was then divided by the number of grid points over the western US domain. The number of timesteps characterized by a fold with

a maximum pressure greater than 400 hPa was also recorded, for a measure of the frequency of tropopause folds under various large-scale conditions. Any mention of tropopause fold frequency therefore refers to only this subset of folds, as folds shallower than 400 hPa are unlikely to affect STT-PBL. Similar results are found when folds larger/deeper than 500 hPa are selected, although differences in the size of intrusions are more difficult to discern given the small-scale nature of intrusions this deep (e.g., Knowland et al. 2017).


We note that the 2-PVU surface, which is used to define the maximum pressure folds reached, does not always mark the terminus of the fold, and that ozone originating in the stratosphere more closely follows the 1-PVU surface which penetrates further downward, as shown in Albers et al. (2018; their Figure 2). Shapiro (1980) also discussed how ozone associated with a tropopause fold in March 1978 reached farther into the troposphere than the dynamic tropopause, indicative of cross-

isentropic mixing. Knowland et al. (2017) examined cross sections of two stratospheric intrusions that led to enhanced surface ozone concentrations in Colorado in spring 2012, and showed that the dynamic tropopause only reached to the mid-troposphere, with only small filaments of the intrusion reaching deeper than 500 hPa, while high stratospheric ozone values extended to the surface. For these reasons, we believe that tracking when the dynamic tropopause is deeper than 400 hPa captures the structures often associated with transport deep into the troposphere. This is confirmed by the strong relationship

between STT-PBL and the fold characteristics discussed in Section 3.3. Changes in these measures of fold depth, size, and frequency will be evaluated during the phases of the spring transition, as well as during El Niño, La Niña and ENSO neutral conditions.



## 2.2 Post-Processing and Diagnostics

We use the leading empirical orthogonal function (EOF1) and principal component (PC1) time series of the daily mean 200-hPa zonal wind over the north Pacific basin (100-280°E, 10-70°N), smoothed with a 5-day running mean, to track the seasonal evolution of the jet. Zonal wind anomalies used for EOF analysis were calculated with respect to the February – June 1958-2017 average using JRA-55 reanalysis (black contours, Fig.1a), with the resultant anomalies intentionally including changes associated with the seasonal cycle (in contrast to the more canonical approach, in which the daily climatology is removed to

eliminate the seasonal cycle). To be consistent with the STT-PBL and tropopause fold datasets, which are based upon ERA-Interim reanalysis (Sprenger et al. 2017), we recomputed the 200-hPa zonal wind EOFs in the same manner using ERA-Interim 200-hPa zonal wind (the correlation between PC1 using JRA55 and ERA-Interim for their common period, 1979-2017, is 0.99).

For a measure of the eddy characteristics and horizontal Rossby wave energy propagation during various phases of the spring transition, the horizontal **E**-vector (Eq. 1; Hoskins et al. 1983) was calculated using daily zonal and meridional wind anomalies that have a) the 60-year daily climatology and b) the 11-day running mean removed. Regions where **E** points eastward (westward) are characterized by meridionally (zonally) elongated eddies (Fig. S2). Negatively tilted anomalies, indicative of cyclonic wave breaking, correspond to a northward-pointed **E** and energy propagation, while positively tilted anomalies

indicate anticyclonic wave breaking and correspond to southward-pointed **E** and energy propagation.

$$\mathbf{E} = [\overline{v'^2 - u'^2}, -\overline{u'v'}] \qquad (1)$$

## 2.3 Significance Testing

For a measure of confidence in the differences in fold characteristics and STT-PBL during different jet phases and ENSO conditions, mean values were bootstrapped using a sample size $N_{eff} = N/t_{autocorr}$, where $t_{autocorr}$ is the number of timesteps at which the autocorrelation of the variables decreases to below 0.5, and N represents the number of samples in the smallest group being compared. For example, when comparing STT-PBL over North America during May El Niño, La Niña and ENSO neutral conditions, with corresponding samples sizes (N) equal to 20, 9 and 7 years, respectively, STT-PBL for the three groups

is resampled using N=7 years* 124 timesteps/year = 868 timesteps, and $t_{autocorr}$ = 6 timesteps (36 hours), which equates to $N_{eff}$ = 144 timesteps. To fairly compare the confidence intervals for each ENSO group, every group was resampled using the reduced sample size $N_{eff}$, to calculate a new mean STT-PBL value. This process was repeated 10000 times to determine the 95[th] and 5[th] percentiles of the mean value for each group. A similar approach was applied to each variable considered.

## 3 Results
### 3.1 Characteristics of the spring north Pacific jet transition



The leading EOF pattern of 200-hPa zonal wind tracks the seasonal evolution of the north Pacific jet from February through June each year (Fig. 1). A positive PC1 value represents the stronger wintertime state (Fig. 1a), which gradually weakens on average from about March through June, as shown by the transition of PC1 from positive to negative each year (Fig. 1b). There is greater spread among the PCs of individual years during February, March and April than there is during May and June, indicating the transition from winter to spring is more variable than the transition from spring to summer. The composite zonal wind on days when PC1 > $1\sigma$, hereafter referred to as the winter phase, most often occurring in February - March, is characterized by a strong jet extending well past the date line (Fig. 2a). During the winter phase, high-frequency eddy kinetic energy (EKE) is greatest in the jet exit region in the central Pacific, representing the wintertime Pacific storm track and tendency for eddies to amplify via deformation in the jet exit region (Rivière and Joly 2006; Breeden and Martin 2018). The prevalence of equatorward-pointed **E**, signifying positively-tilted waves, over North America is consistent with the frequency of positively-tilted troughs and ridges identified during boreal winter by Schemm and Sprenger (2020).

As PC1 decreases to values between +/- $.5\sigma$, which we define as the transition phase, the jet core weakens substantially while the jet exit region shifts northward (Fig. 2b). The storm track is more energetic throughout the Pacific-North American region compared to the winter phase, and shifts northward with the jet exit region. In the east Pacific, a distinct secondary jet maximum develops near Hawaii in the subtropics, creating a double-jet structure in the Pacific basin which differs substantially from the strong, merged wintertime jet. The formation of this secondary zonal wind maximum was also observed to develop in April by Newman and Sardeshmukh (1998). The magnitude of **E** increases during the transition phase, with meridionally-elongated, positively-tilted waves dominating the structure in the midlatitude Pacific. Such characteristics are related to frequent anticyclonic Rossby wave breaking associated with the formation of the two jet maxima observed in the transition phase. A distinct region of nearly-zonal **E** is observed over the eastern Pacific/western US, indicating waves in this region are more amplified meridionally compared to when the jet occupies the winter phase. Zonal wind, EKE and eddy amplitude proceed to weaken by late spring/early summer, when PC1 < -$1\sigma$, hereafter referred to as the summer phase (Fig. 2c), with the subtropical jet in the eastern Pacific essentially disappearing altogether. Over the western US, eddies are still meridionally amplified but less so than during the transition phase, characteristic of the weakened storm track.

To examine the variability in the spring transition, we tracked the date on which PC1 dropped below +$.5\sigma$ and remained below that value for the remainder of the season. We target this transition in particular given the high variability of PC1 early in the season, and the marked invigoration of the storm track associated with PC1 decreasing from strongly positive to neutral. The mean transition date over the 60-year record is 4 April, with a standard deviation of +/- 12 days. To test if there are dynamic differences in the transition if it occurs earlier or later than normal, we grouped each season into early, neutral and late transition years, requiring early (late) transition years to have a transition date at least 5 days earlier (later) than the 60-year average (Table 1). The average timing of the transition for the three groups differs by about two weeks, with the early group



transitioning on average in mid-March, the late group in mid-April. Comparing the composite February – June evolution of PC1 for the early and late groups (the neutral transition years fall in-between), PC1 in the early group begins to decrease near the beginning of March, although these differences are not significant until later in the month (Fig. 3). During April, the late group PC1 value is roughly .5σ higher than the early group, an average zonal wind difference of 10 m s$^{-1}$ within the jet core, while by May the two groups are indistinguishable from one another. An early winter-to-spring transition is not associated

with an early spring-to-summer transition, with PC1 decreasing below -.5σ in mid-May for all transition groups. To test whether early transitions are more abrupt (and therefore more dynamically disruptive) than later transitions, we compared the composite evolution of PC1 with respect to each year's transition date, and did not find any significant differences in the vigor of the transition (Fig. S3).

**3.2 Relationship between the Spring Transition and STT-PBL**

This section will show how the spring transition modulates STT-PBL over western North America. Earlier transitions enhance the amount of the time the jet occupies its transitional phase, corresponding to a more invigorated storm track, more folds and therefore more STT-PBL than later transitions. Early in the season, *deeper* folds enhance STT-PBL, while later in the season

more *expansive* folds enhance STT-PBL.

STT-PBL is modulated by the phase of the jet and corresponding invigoration of the storm track (Fig. 4). Transport increases by roughly threefold when the jet is in its transition phase, compared to the composite STT-PBL during both the winter and summer phases. STT-PBL was averaged over western North America (box in Fig. 4b) for each day in the record and

subsequently binned by PC1, confirming STT-PBL is strongest when the jet is closer to its transitional phase than at either extremity (Fig. 5a; Fig. S4). Both the highest STT-PBL days in the record and the highest median STT-PBL values occur during the transition phase, while the distributions during the winter and summer phases are indistinguishable from one another (Fig. 5a). Consistent with the STT-PBL changes, tropopause folds reach farthest into the troposphere during the transition phase, on average to 450 hPa, in contrast to median values near 400 hPa during the winter phase and 300 hPa during the

summer phase (Fig. 5b). During the winter phase, shallow (<300 hPa) and deep (>400 hPa) folds are equally likely, while deep folds are more frequent than shallow folds during the transition phase. Shallow folds are overwhelmingly more likely during the summer phase, which might be related to the weaker jet and associated ageostrophic circulation during summer. Thus, while the STT-PBL distributions during the winter and summer phases are indistinguishable from one another, the fold depth distributions differ substantially.


In addition to changes in tropopause fold depth during the spring transition, the daytime BLH in the interior West increases dramatically, meaning shallower folds can reach the top of the boundary layer. Consistent with the STT data, 6-hour forecasts of PBL height valid at 1800 UTC were averaged over western North America and grouped by jet phase (Fig. 5c), confirming



the PBL deepens as the jet transitions. Thus, while folds deeper than 500 hPa still occur somewhat frequently during the winter
phase, the PBL is far lower, meaning a smaller subset of folds is deep enough to penetrate the boundary layer compared to the
transition phase. Conversely, when the jet occupies the summer phase, despite a very deep boundary layer, there is limited
transport due to a relative lack of intrusions deeper than 350 hPa. The transition phase is associated with higher STT-PBL
through the fortuitous coincidence of both more frequent deep tropopause folds *and* a deepening PBL. We note that STT-PBL
can be displaced spatially from the position of the tropopause-level folds measured here, and can be aided by lower-
tropospheric vertical motions such as those occurring around frontal zones and convection (Skerlak et al. 2019, their Figure
1).

Reconsidering the eddy characteristics associated with the three jet phases (Fig. 2), it appears that tropopause folds deep enough
to produce STT-PBL occur most often when waves are highly amplified and the storm track is most energetic. Highly amplified
Rossby waves are associated with strong curvature, particularly on the western edge of troughs, producing the subsidence that
forms deep tropopause folds and STT-PBL (e.g., Sprenger et al. 2007). Amplified waves propagating over western North
America, which occur most often during the transition phase, bring more folds over the high terrain of the Rocky Mountains
as the PBL deepens, leading to the STT-PBL maximum observed in boreal spring.

Given the longer period of time the jet is within its transition phase (when PC1 +/- 0.5σ), hereafter referred to as the residence
time, (Fig. 3), we hypothesize that early transition years are characterized by more STT-PBL than late transition years. To that
end, we compared monthly mean STT-PBL for the early and late transition groups, revealing there is indeed more STT-PBL
during early transition years in March, April and May, coinciding with more frequent folds deeper than 400 hPa (Fig. 6a-b).
In February and March, folds are deeper and larger in early transition years as well, while in later months folds are larger but
not deeper (Fig. 6c-d). The residence time of the jet is much greater during early years by definition, and monthly mean EKE
over the North Pacific (180-250°E, 40-60°N; box in Fig. S6), is greater in April during early years (Fig. 6e-f). Compositing
each variable with respect to each year's transition date reveals an upward shift in STT-PBL in the two weeks following the
transition in both groups, coincident with a marked increase in tropopause folds, residence time and EKE (Fig. 6g,h,k,l). Fold
depth and area, conversely, are not systematically affected by the transition (Fig. 6i,j).


The relationship between STT-PBL and each related but distinct fold characteristic – maximum depth, fold area and frequency
– evolves over the course of the spring transition. This is evident from timeseries of the correlation between monthly mean
STT-PBL, February – June, with each fold characteristic, residence time of the jet, and median PBL height (Fig. 7; Fig. S4).
During February and March, STT-PBL has the strongest correlation with fold depth and frequency, consistent with intuition.
In April, however, the relationship between STT-PBL and fold depth diminishes, while fold frequency maintains a strong
relationship with transport through June. In contrast to fold depth, the relationship between fold size and STT-PBL is strongest
in May, when it has the second-strongest relationship after frequency. A longer residence time of the jet within the transition





phase enhances STT-PBL in March and February, a relationship which disappears later in spring, in part because PC1 continues decreasing towards its summertime state. Daytime PBL height and STT-PBL are modestly correlated in February and March,
with no relationship in later months when the daytime PBL has deepened to several kilometres and appears to no longer be the limiting factor for deep transport (Fig. 5c; Seidel et al. 2012; Langford et al. 2017). Since fold frequency maintains a strong relationship with STT-PBL throughout the transition, we correlated fold frequency to EKE, confirming a more energetic storm track produces more folds and supporting the relationship between storm track variability, folds and STT-PBL. The correlation drops off by May, however, for reasons which are not immediately clear but might reflect the more convective nature of
transport during this time of year, which can be important for transport to the surface (Langford et al. 2017; Skerlak et al. 2019). Overall, fold frequency maintains the strongest relationship with STT-PBL throughout the transition, while fold depth and area also affect STT-PBL early and late in the transition, respectively.

**3.3 Impact of ENSO on the Spring Transition and STT-PBL**

What drives the substantial variability in the timing of the spring transition? While prior research has alluded to a connection between ENSO and STT-PBL, the precise nature of the ENSO-fold-STT relationship during boreal spring is not fully understood. Here we demonstrate that ENSO conditions do influence the jet, tropopause fold characteristics and STT-PBL, and that this influence evolves throughout the spring transition.

ENSO markedly affects the jet from February – April, with La Niña conditions corresponding to a much lower PC1 value than neutral or El Niño conditions, while in May and June the differences are weaker (Fig. 8). There is some asymmetry in the PC1 response, with La Niña weakening the jet more substantially than El Niño strengthens it. Given the positive relationship between STT-PBL and residence time of the jet, we hypothesize that La Niña conditions are associated with enhanced STT-PBL, which is broadly confirmed in Figures 9-10 and is consistent with the conclusions of Lin et al. (2015). In February and
March, El Niño conditions are associated with a zonally extended jet that connects to the jet over North America, while the jet is zonally confined to the central Pacific during La Niña (Fig. 9a-f). STT-PBL is overall weak in February, but is strongest during La Niña years, consistent with the most neutral PC1 value. STT increases in all three ENSO groups during March, as PC1 values decrease. Note that the jet has already transitioned during many of the La Niña and some of the ENSO neutral years (Table 1). In April, the jet transition is either underway or has already occurred, and correspondingly STT-PBL peaks
for the El Niño and ENSO neutral groups, and remains elevated for La Niña (Fig. 9g-i; Fig. 10a). During May and June, STT-PBL remains elevated during La Niña years, although the difference compared to ENSO-neutral is somewhat uncertain with the number of samples available (Fig. 9j-o). The (presumably eddy-driven) jet core in the western Pacific is stronger during La Niña years, reflecting an increase in storm track activity compared to neutral and El Niño conditions coincident with a more negative PC1 value (Fig. 10f; Fig. S6).






Which of the various tropopause fold characteristics explored in the prior section do ENSO conditions affect? Just as the influence of ENSO on PC1 evolves over the course of the transition, so too does the influence of ENSO on tropopause folds. During February and March, La Niña conditions are characterized by significantly deeper and more frequent folds, driving an increase in STT-PBL (Fig. 10a-c). Folds are also larger, particularly in May when the relationship between fold area and STT-PBL is the strongest (Fig. 10d). While STT-PBL during April is similar in all three groups, folds are still more frequent and potentially deeper during La Nina (Fig. 10b-c). STT-PBL is elevated during La Niña in May, when folds are more common and larger in areal extent. Mean fold depth, in contrast to fold size, is insensitive to ENSO phase in May and June. Finally, the residence time of the jet within its transition phase is significantly enhanced in February and March (as suggested in Fig. 8), while it is reduced in May when PC1 is more negative (Fig. 10e; Fig. 9l). EKE during La Niña is most enhanced in April, similar to the early transition years (Fig. 10f; Fig. 6f), due to the notable increase in EKE following the transition (Fig. 10l; Fig. 6l). Note that during May, EKE is enhanced during La Niña in a smaller region over the eastern Pacific (Fig. S6d), accompanied by a more zonal **E**-vector, consistent with elevated STT-PBL observed in Fig. 10a but not reflected in Fig. 10f.

In summary, in late winter/early spring, the teleconnection to the extratropics during La Niña projects onto the seasonal transition of the jet represented by PC1, often expediting the transition. This large-scale modulation, in turn, enhances the depth of tropopause folds and fold frequency over western North America, enhancing STT-PBL. In May, La Niña conditions continue to increase the frequency and size of folds and therefore STT-PBL, also through invigoration of the storm track as in February and March (Fig. S6). An opposite response is observed during El Niño conditions.

## 4 Discussion and Conclusions

The present study seeks to further clarify the relationship between the north Pacific jet, tropopause folds, and deep mass transport, and how these connections evolve from February – June over the western United States using JRA-55 and ERA-Interim reanalysis. The leading EOF and corresponding PC of 200-hPa zonal wind are demonstrated to track the winter-to-summer jet evolution. The nature of this transition is consistent with previous studies of the annual cycle of the jet (Newman and Sardeshmukh 1998), and the associated changes in the storm track (Nakamura 1992; Hoskins and Hodges 2019). We find that the spring jet transition modulates folds and STT-PBL, and that the timing of the transition varies from mid-March to late April. In February and March, early transitions lead to enhanced STT-PBL through an increase in the depth and frequency of tropopause folds over western North America. Conversely, late transitions are characterized by shallower, less frequent folds and weaker STT-PBL. Early transitions preferentially occur during La Niña conditions, while there is a weaker but still notable link between El Niño conditions and late transitions. In February and March, La Niña conditions enhance STT-PBL through an increase in fold *depth* and frequency, while in May, STT-PBL is greater through an increase in fold *size* and frequency.

The peak in STT-PBL during boreal spring over western North America found by previous studies occurs through the simultaneous occurrence of the dynamic north Pacific jet transition and seasonal deepening of the PBL. The highly amplified





flow observed during the spring transition increases the frequency of deep stratospheric intrusions, as the PBL deepens due to enhanced solar heating, strengthening STT-PBL. The association between more STT-PBL and highly amplified flow found here is consistent with case studies of notable stratospheric ozone intrusion events over the western US (Langford et al. 2009; Lin et al. 2015; Knowland et al. 2017) and the established role of Rossby wave breaking in facilitating STT (Homeyer and Bowman 2013). The zonal wind anomalies associated with the transitional phase also resemble the April-May zonal wind

anomalies found during years with the greatest mixing ratios of ozone observed in stratospheric intrusions (Albers et al. 2018). The present analysis offers a simple metric to track such jet variability and situates it within the context of the seasonal transition.

Our results are consistent with the differences in STT-PBL of ozone observed between ENSO phase during April-May over

western North America by Lin et al. (2015). This is notable because we only consider mass transport without measuring how the ozone concentrations within folds varies between ENSO, which can be quite substantial (Garcia-Herrera 2006; Neu et al. 2014; Albers et al. 2018). The influence of ENSO on the ozone reservoir opposes the effect of ENSO on folds - namely, La Niña (El Niño) conditions reduce (enhance) extratropical lower stratospheric ozone concentrations, by modification of the Brewer-Dobson Circulation (Neu et al. 2014, Albers et al. 2018). As such, from our results alone drawing conclusions about

deep ozone transport are limited, while deep mass transport is clearly modified by ENSO. Finally, we note that STT-PBL does not necessarily guarantee transport of ozone or mass all the way to the surface, which can be strongly influenced by PBL dynamics and ageostrophic circulations around low-level frontal zones (Skerlak et al. 2019).

This study took advantage of recently-developed products specifically targeted at understanding STT-PBL using ERA-Interim

reanalysis fields (Sprenger et al. 2007; Dee et al. 2011; Skerlak et al 2014; Sprenger et al. 2017). We note that, as a consequence, our results concerning STT-PBL are limited to the ERA-Interim record and the frequency of ENSO events within the 1980-2016 period (excluding section 3.1 which used the 60-year JRA-55 reanalysis record). To minimize the possibility of overstating subsequent conclusions regarding folds and STT-PBL, we have applied rather strict significance testing to account for sampling and autocorrelation, confirming the differences we have highlighted are frequently statistically

significant. Future work could employ model simulations using many ensembles to increase the sample size of early/late transition years and ENSO events, to revisit the connections found in this study.

**Code and Data Availability**

The code used to perform this analysis can be accessed by personal communication with the corresponding author. The

reanalysis products used in this study are available through the National Center for Atmospheric Research / University Consortium for Atmospheric Research Data Archive: https://rda.ucar.edu/. The tropopause fold metrics and STT-PBL variable is available at monthly resolution from http://eraiclim.ethz.ch/.



**Author Contributions**

Melissa L. Breeden wrote the code, produced the figures and wrote the manuscript. Amy H. Butler and John. R. Albers provided computational resources and frequent guidance during the study, and provided edits/comments to the manuscript. Michael Sprenger provided the STT-PBL and tropopause fold datasets and provided edits/comments to the manuscript. Andy O. Langford provided edits and comments to the manuscript.

**Competing Interests**

The authors declare no conflict of interest.

**Acknowledgements**

The authors would like to thank Dr. Matthew Newman for constructive conversations regarding the spring transition, which
in part motivated this study. This research was supported by the NOAA Climate and Global Change Postdoctoral Fellowship Program, administered by UCAR's Cooperative Programs for the Advancement of Earth System Science (CPAESS) under award # NA18NWS4620043B.

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

**Tables**

| Early Transition Years Mean Transition Date: March 21 | On Time Transition Years Mean Transition Date: April 3 | Late Transition Years Mean Transition Date: April 17 |
|---|---|---|
| 1960 (March 22) | **1958 (March 30)** | **1966 (April 16)** |
| 1967 (March 25) | **1959 (March 30)** | 1970 (April 13) |
| 1968 (March 20) | 1961 (April 6) | *1975 (April 9)* |
| **1969 (March 28)** | 1962 (April 3) | 1978 (April 15) |
| *1971 (March 25)* | 1963 (March 30) | **1983 (April 11)** |
| 1972 (March 23) | 1964 (April 2) | **1987 (April 14)** |
| *1976 (March 28)* | 1965 (April 4) | 1990 (April 21) |
| 1977 (March 28) | 1973 (April 3) | **1992 (April 22)** |
| 1982 (March 26) | *1974 (April 3)* | 1993 (April 26) |
| *1985 (March 4)* | 1979 (April 4) | 1995 (April 17) |
| *1989 (March 28)* | 1980 (April 3) | 1996 (April 18) |
| 1991 (March 25) | 1981 (March 30) | 1997 (April 25) |
| *1999 (March 9)* | 1984 (April 4) | 2004 (April 15) |
| *2000 (March 23)* | 1986 (April 6) | 2005 (April 30) |
| 2002 (March 28) | 1988 (April 1) | 2007 (April 19) |
| *2008 (March 16)* | 1994 (April 8) | 2013 (April 19) |
| 2010 (March 27) | **1998 (April 5)** | 2014 (April 16) |
| *2012 (February 27)* | 2001 (April 5) | **2016 (April 13)** |
|  | 2003 (April 2) | 2017 (April 9) |
|  | 2006 (April 1) |  |
|  | *2009 (April 4)* |  |
|  | *2011 (April 5)* |  |
|  | **2015 (April)** |  |

**Table 1: Years when the jet transitioned early, on time, or late, relative to the mean transition date of April 3. Italic**
**(bold) text denotes years when La Niña (El Niño) conditions were observed the month of the transition date.**





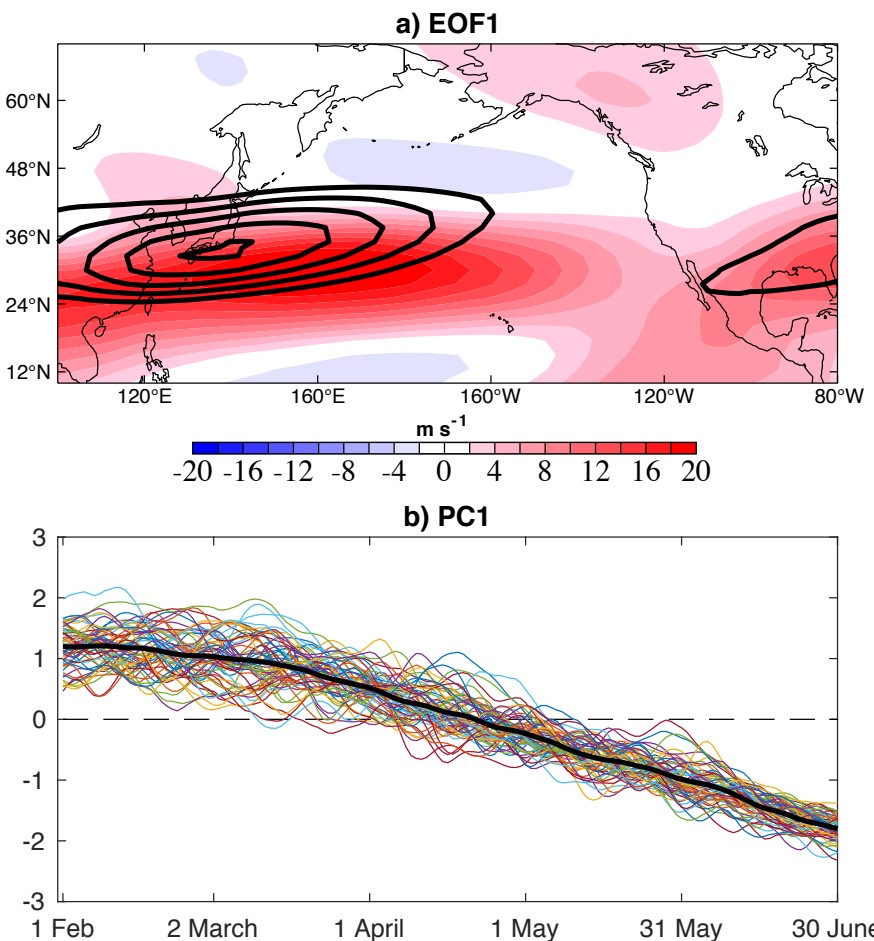

**Figure 1: The color shading in a) is the EOF1 zonal wind anomaly pattern associated with a +1σ PC1, and the black contours show the FMAMJ 1958-2017 mean zonal wind, contours starting at 30 m s⁻¹ every 5 m s⁻¹. The percent variance explained by EOF1 is 40% and is well separated from the next EOFs according to the criterion of North (North 1982). b) The thin lines plot each year's PC1 evolution, and the thick black line shows the 60-year average.**




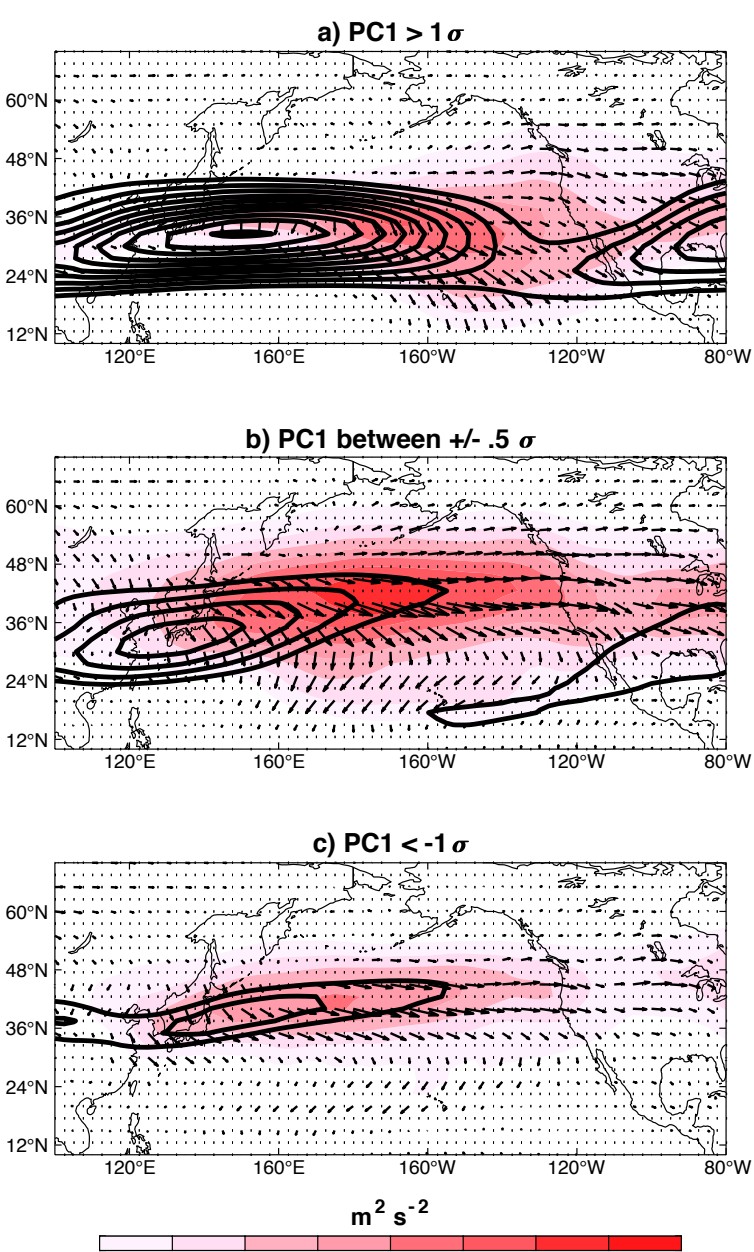

**Figure 2: The black contours show the full-field 200-hPa zonal wind composited for days characterized by a) positive PC1 (N= 1720), b) neutral PC1 (N=1455) and c) negative PC1 (N=1855). Zonal wind contours start at 30 m s⁻¹ at intervals of 5 m s⁻¹. The color shading shows the composite high-frequency EKE during the positive, neutral and negative jet phases, and the corresponding composite horizontal E-vector is shown in the black arrows.**





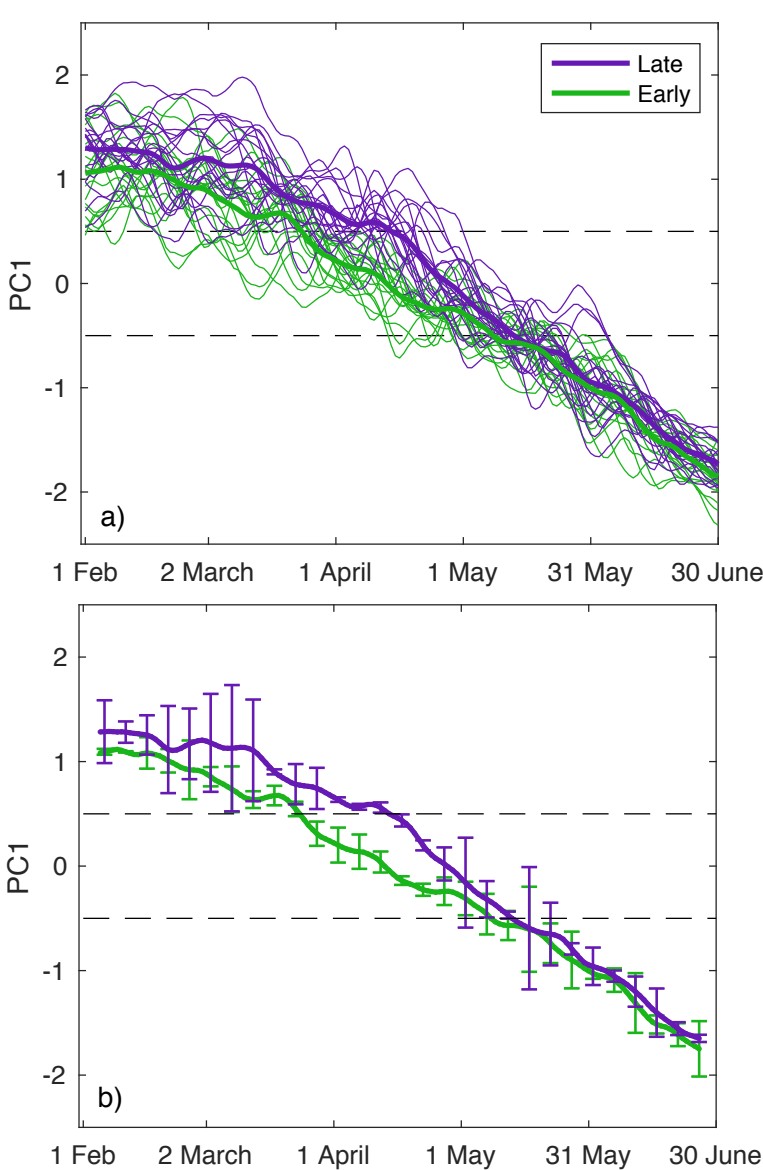

**Figure 3: a) The thick green and purple lines show the composite PC1 evolution, and the thin green and purples lines show each year's PC1 evolution, during 18 early and 19 late transition years, respectively. b) shows confidence intervals for the composite PC1 values for the early (green) and late (purple) groups.**





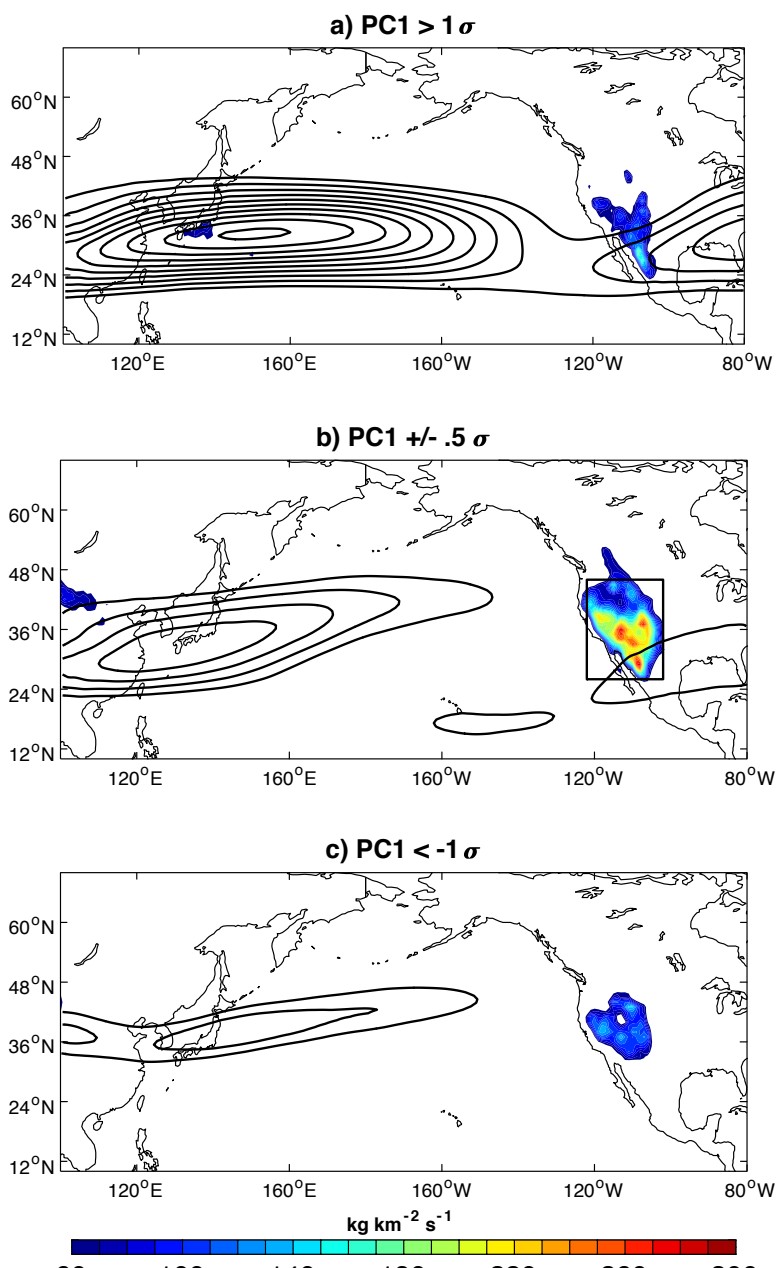

**Figure 4: Composite STT-PBL during the three jet phases using ERA-Interim PC1 for 1980-2016. Top shows STT-PBL during days characterized by a wintertime value (N=998), middle shows STT-PBL during days characterized by a transitional PC1 value (N=952) and bottom shows STT-PBL during days characterized by a summertime PC1 value (N=1125). Zonal wind contours start at 30 m s⁻¹ at intervals of 5 m s⁻¹.**



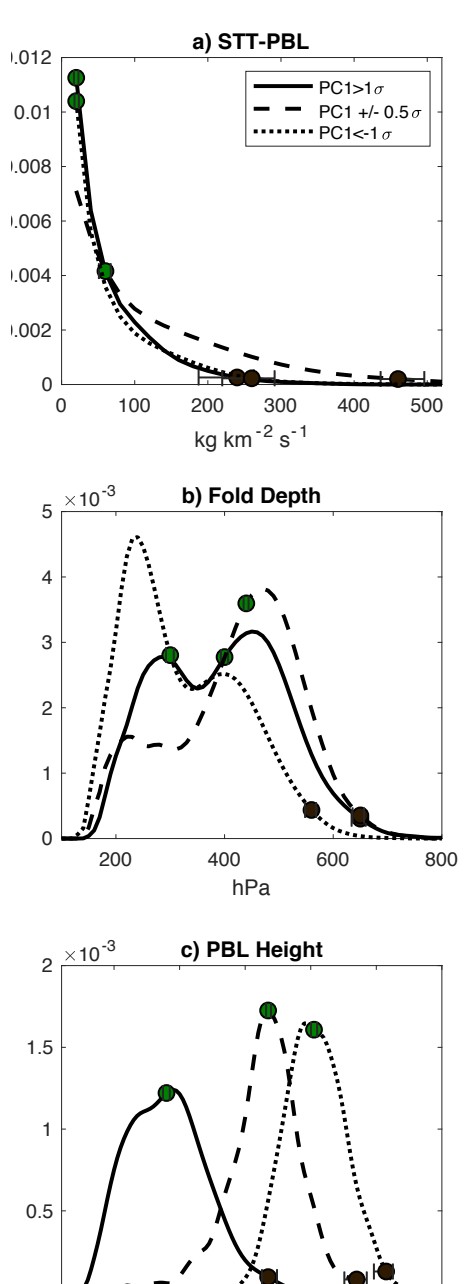

**Figure 5: a) Distribution of STT-PBL during positive (solid), neutral (dashed) and negative (dotted) jet phases. b) as in a) but for the distribution of the maximum tropopause fold pressure observed within folds passing over the western United States. c) As in a) but for 6-hour forecasts of 1800 UTC PBL height. The green dots show the bootstrapped medians of each distribution, and the black dots show the bootstrapped 99th percentile values.**



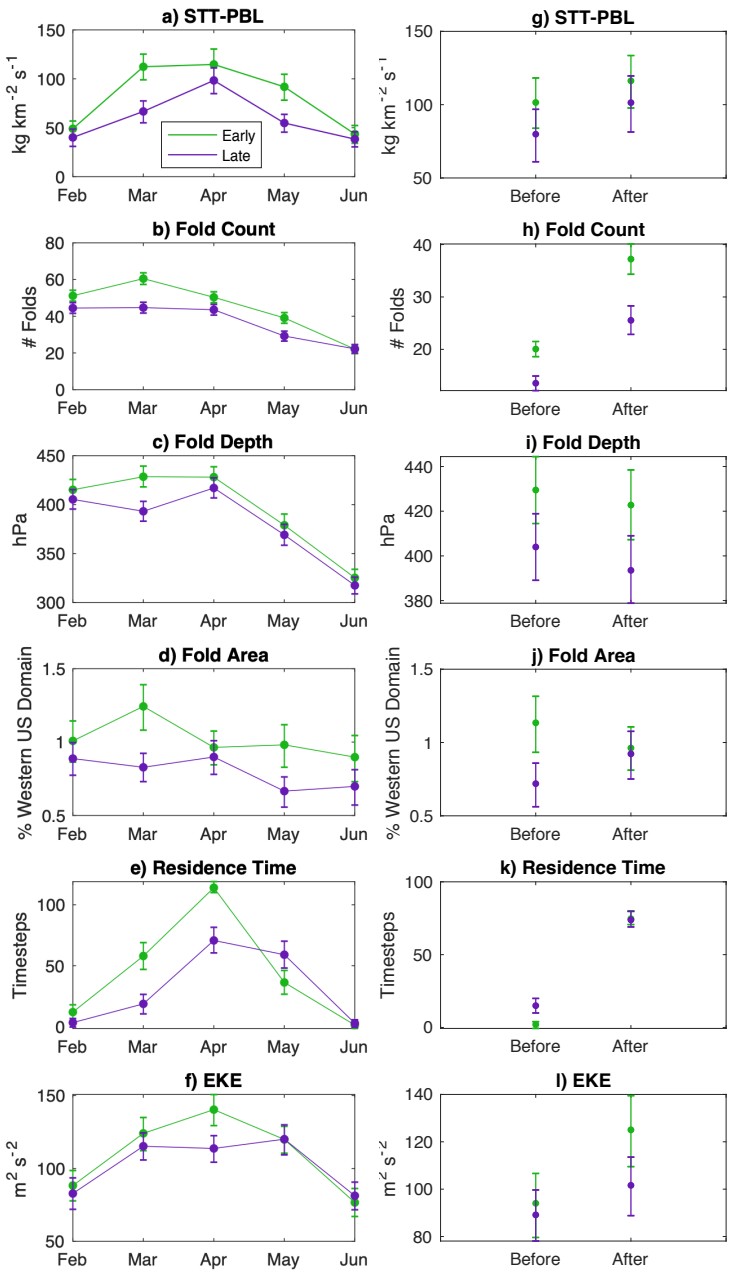

**Figure 6: Bootstrapped a) mean STT-PBL, b) fold frequency, c) fold depth, d) fold size, e) mean residence time of the jet and f) mean EKE for February – June during early transition years (green) and late transition years (purple). g) – l) show the same variables averaged over the period 5 to 20 days prior to each year's transition date (Days -20 to -5; Table 1), labelled 'Before', and the two weeks following the transition date, Days 0 to +14, labelled 'After'.**





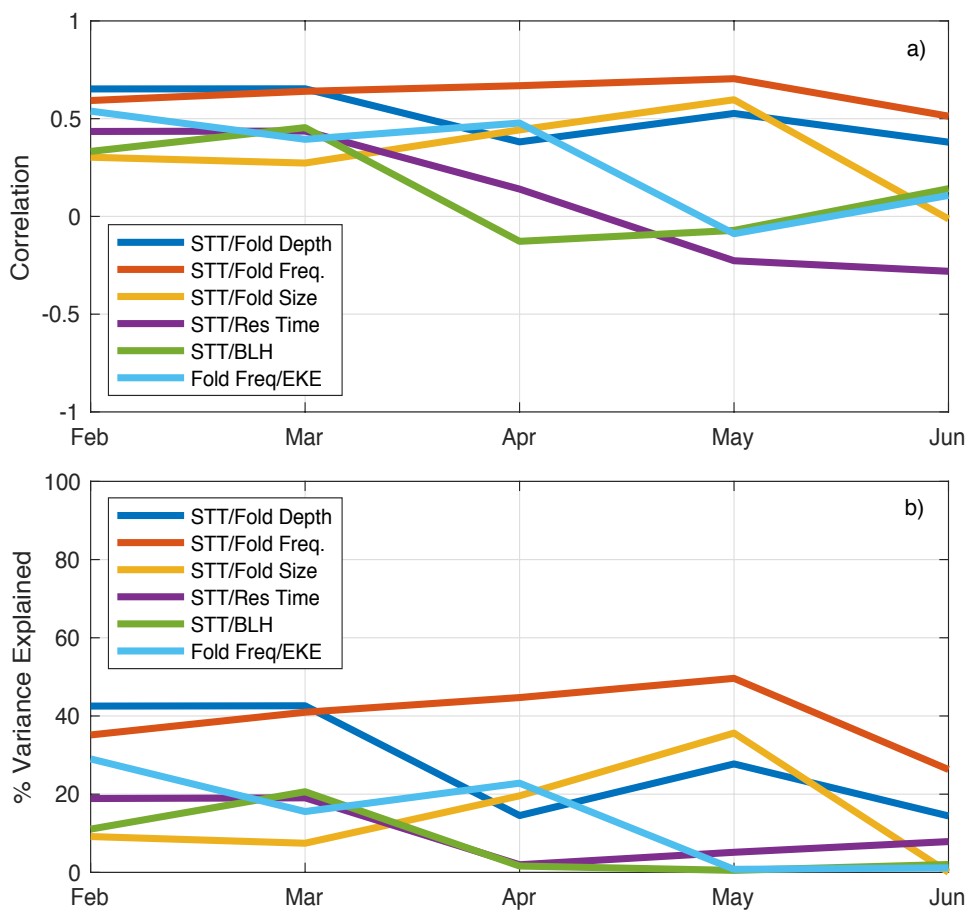


**Figure 7: a) Time series of the correlation (r) of monthly mean STT-PBL with various tropopause fold characteristics, residence time of PC1 +/- .5, mean PBL height and mean EKE. b) shows the corresponding percent variance (r$^2$) explained by each relationship.**


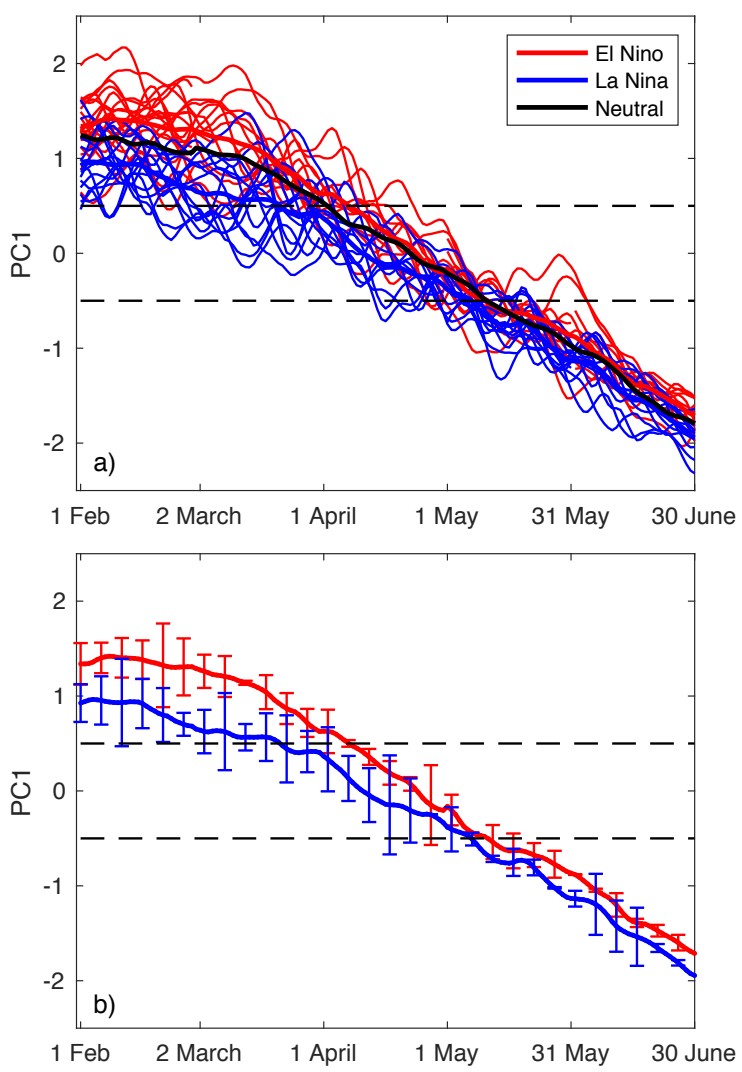

**Figure 8: a) The thin lines show PC1 values during months of La Niña events (blue) and El Niño events (red), with the thick blue (red) line showing the average La Niña (El Niño) periods. The thick black line shows the PC1 average for the remaining neutral months. b) shows the bootstrapped confidence intervals for the mean PC1 values for La Niña events (blue) and El Niño events (red).**






**Figure 9: The color shading shows the monthly mean STT-PBL during El Niño conditions (left), ENSO neutral conditions (middle) and La Niña conditions (right). The black contours show the monthly mean 200-hPa zonal wind. The mean PC1 values for each month and ENSO group are shown in the bottom left corner. Zonal wind contours start at 30 m s⁻¹ at intervals of 5 m s⁻¹.**


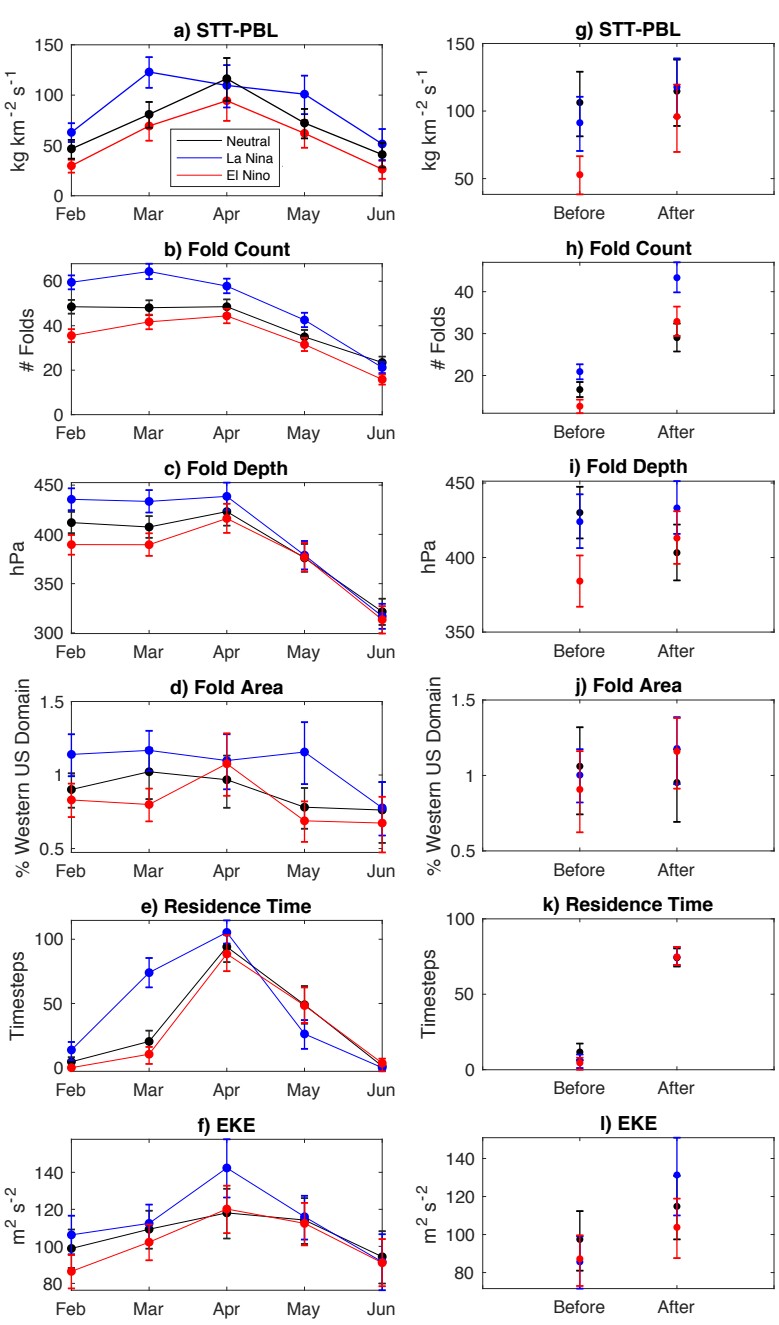

**Figure 10: As in Figure 6 but grouped by El Niño years (red), La Niña years (blue) and ENSO neutral years (black).**