# Peer review of "The Spring Transition of the North Pacific Jet and its Relation to Deep Stratosphere-to-Troposphere Mass Transport over Western North America"

_Atmospheric Chemistry and Physics, 2020_

## Referee Comment (RC1) · Anonymous Referee #2 · 22 Oct 2020

This is one of the most polished papers I have reviewed in several years! The investigation is comprehensive, well motivated, and the text and figures are of high quality. It is an excellent contribution to the field and was a pleasure to read. Many thanks to the authors for the careful and considerable work put into this manuscript! I have only a handful of comments/suggestions outlined below.

Lines 41-44: It seems that an annual cycle of tropopause altitude would also be relevant here. Perhaps it frequently reaches a minimum in late winter/early spring.

[Figure]

Line 82: Why is 2.5°Âăresolution data used instead of finer data that is available? It seems reasonable to me since JRA-55 is used mainly for context, but a simple justification along those lines would be nice to include.

Lines 116-127: The mention of "dynamic tropopause" here is sudden. I would recommend simply specifying the range of PV thresholds used in these studies. Otherwise, the discussion could become confusing and is a bit nuanced as is.

Line 180: I recommend citing Rossby wave breaking studies supporting this statement.

Line 307: I would recommend also pointing out that the ENSO-jet relationship is consistent with Rossby wave breaking activity given the occurrence of the jet breaks during La Niña.

Line 339-347: Excellent points here! :)

―――――――――――――――――

---

## Referee Comment (RC2) · Anonymous Referee #1 · 23 Oct 2020

This study presents the results of a thorough statistical analysis of springtime stratosphere-to-troposphere transport to the boundary layer (STT-PBL) over the western USA, in the context of the jet structure transition between the winter and summer regimes. The authors use wind fields from the ERA-Interim and JRA-55 reanalyses to identify the dominant Pacific wind patterns at the jet level by the way of EOF analysis and employ previously developed methods to calculate STT-PBL and associated diagnostics. They demonstrate that the intensity of springtime STT-PBL is a function of the timing of the jet transitions, with early transitions leading to more intense STT-

PBL driven by deeper and more frequent tropopause folds. Furthermore, they show that the transition timing is correlated with ENSO, and carefully investigate the mechanisms involved, by performing a simultaneous analysis of Rossby wave propagation diagnostics, tropopause folds diagnostics, and PBL depth distributions. The paper is really well written, logically constructed and easy to follow, which is no small feat given the complexity of the subject. It also does a great job referencing the sizable body of literature on the topic. I think the analysis and conclusions are very solid and the paper meets all the ACP criteria for publication. Despite my best efforts (it's my job after all;)) I couldn't find almost any issues with the analysis or presentation. It's a great and important paper and I enthusiastically recommend it for publication almost as is. I do have a few very minor suggestions for edits. The most important one concerns the methods section that, I think, would benefit from adding some more details (even if those details can be found elsewhere) that would make the paper more self-contained.

Minor and technical comments

L82-86. What's the vertical resolution of each reanalysis and why is it sufficient for the present purposes, particularly for driving the trajectory model?

LL85-89. Can you comment on how the changing observing system (pre-satellite to satellite to more satellites) in JRA-55 impacts the results?

L91-93. Can you expand this paragraph a bit? This is one of the main tools used here, and I found it hard to get even a general idea of what's being done there without reading Skerlak et al. 2014. For example, how are the trajectories calculated? Perhaps a simple diagram in the supplementary material?

LL 116-118. Is it possible that this is resolution dependent? Can you comment on that?

LL207-210. I like the idea of stating the main results in the first concise paragraph. However, at the initial pass, it wasn't clear to me if the second and third sentences are the results or something that we already know. How about something like "The main

findings are:..." after the first sentence?

L220. I'm guessing the bimodality in 5b arises from an oscillation between the two phases during the transition period. It's pretty neat. Can you add a one/two-sentence comment on that?

L275. Variability reflected in Fig. 3, right? If so, can you reference Fig. 3 explicitly?

LL344-345. I'm struggling to understand this sentence. Please rephrase; it looks like an important point is being made there.

Fig. 3 caption. The dashed lines are +/-1 sigma, right?

---

## Short Comment (SC1) · 2 Nov 2020

This review was prepared as part of graduate program course work at Wageningen University, and has been produced under supervision of Prof Wouter Peters. The review has been posted because of its good quality, and likely usefulness to the authors and editor. This review was not solicited by the journal.

The paper by Breeden et al. (2020) explores the relationship between the spring transition of the north Pacific jet and stratosphere-to-troposphere mass transport to the plan-

etary boundary layer (STT-PBL) over western North-America. Additionally, the spring transition is linked to the state of the El-Niño Southern Oscillation (ENSO). Analyses are based on the JRA-55 and ERA-interim reanalysis datasets. Interannual variability in the spring peak in STT-PBL found in previous studies is shown to relate to the timing of the spring transition, with larger values of STT-PBL for earlier transitions. Finally, ENSO is found to modulate this timing, with earlier transitions being more prevalent for La-Niña conditions and vice versa. This study adds to previous research by providing underlying mechanisms behind the (variability in the) spring peak in STT-PBL over western North-America that was found before. These results suggest that STT-PBL strength can be predicted based on knowledge of the north Pacific jet and ENSO state in the preceding months. This is relevant for, for instance, air quality prediction at the surface as STT-PBL can function as a natural source of ozone in the PBL. In general, the paper is well-written; the structure and headers of the paper are clear and help to understand the research, the figures diversely visualize the results in both maps and time series, the physical mechanisms discussed in the paper are consistently explained well and create a logic story and the results are frequently discussed in light of previous research. Moreover, I think the paper fits nicely in the scope of this journal. Despite the relatively local study area, the implications of this study are thought to be generally applicable in atmospheric sciences and can therefore be extended to other locations around the world that show similar dynamical patterns. Nevertheless, a significant revision of this paper is required before it can be accepted for publication in my opinion.

Most importantly, in the calculation method of STT-PBL described in line 91-96 (section 2.1) of the paper I miss an assessment of the uncertainty in the calculation that is associated with the sensitivity to parameter choices. Previous studies have shown that this sensitivity might not be insignificant, so that the specific parameter choices could substantially affect the outcome of the results in section 3.2 and 3.3 of this study. Firstly, Holton et al. (1995) show that the 380 K potential temperature surface used in the paper by Breeden et al. (2020) as (one of) the definition(s) of the tropopause

coincides relatively well with the tropopause in the tropics, but that this value drops for higher latitudes to approximately the 340/350 K potential temperature surface in the region that is considered in the paper (western North-America). Secondly, Skerlak et al. (2014) showed that their STT(-PBL) calculation is highly sensitive to the choice of minimal residence time of the air in the stratosphere/troposphere before/after a crossing of the tropopause. They find that this sensitivity can be very well approximated by a power law with an exponent of -0.5, which means that the STT estimates for a residence time of 24 and 48 hours respectively deviate by as much as 30 percent. The choice is made by Skerlak et al. (2014) to use a constant value of 48 hours for this parameter. This is validated based on the fact that, although the calculated values of STT-PBL are highly sensitive to the parameter value, the geographical distribution of STT-PBL in which the authors of that study are interested is not. However, in the study of Breeden et al. (2020), this validation does not hold anymore as the STT-PBL calculation is applied to the specific region of western North-America instead of it being used to assess the geographical distribution. Thirdly, the STT-PBL calculation also depends on the accuracy of the PBL height forecast, as this influences the number of trajectories that reach the PBL from the stratosphere. To determine the PBL height the critical Richardson number value of 0.25 is used as the criterion for the PBL top (i.e. the transition from turbulent to laminar flow at the top of the PBL). Troen and Mahrt (1986) indicate that this critical value used for the Richardson number does not have a large influence on the PBL height estimation in unstable conditions, but that it does induce variability in PBL height for neutral conditions. Furthermore, according to Seidel et al. (2012) PBL height is especially uncertain over areas with high elevation, which is the case for parts of the study area of the paper by Breeden et al. (2020) due to the presence of the Rocky Mountains. Altogether, I strongly advise including a sensitivity analysis of the results of section 3.2 and 3.3 to the choice of the parameter values used in the calculation of STT-PBL in order to assess the robustness of the current conclusions. I suggest this sensitivity analysis to include (based on the above discussion) the potential temperature surface that is taken to represent the tropopause,

the minimum residence time of the trajectories that contribute to STT-PBL and both the forecast uncertainty and the forecast value of the PBL height (as a function of the critical Richardson number chosen to represent the top of the PBL).

Additionally, the rationale of using of Japanese Reanalysis-55 dataset does not become clear to me from the paper. In line 85-86 (section 2.1) the authors mention that this dataset is used because of its relatively long record of ENSO events. Yet, the JRA-55 dataset is only used for assessing the characteristics of the spring transition in section 3.1, as is stated in line 352, without considering any influence of ENSO. According to line 86-87 this is because the transport and tropopause fold diagnostics are derived from the ERA-interim reanalysis instead of the JRA-55 data and, therefore, the former is to be used for the analysis of the relationship between ENSO and the spring transition and STT-PBL in order to be consistent in the data used. Therefore, I would like to ask the authors what the exact benefit of using the JRA-55 dataset is and to incorporate the explanation of this in the description of the data in section 2.1. Additionally, it seems to me that the JRA-55 dataset can in fact be used in the analysis of the spring transition for the different ENSO states in section 3.3 (figure 8) as this analysis does not concern any mass transport or tropopause fold characteristics yet and table 1 shows that data on the ENSO states during the spring transition is available for the JRA-55 dataset. Therefore, I would suggest using the JRA-55 dataset instead of the ERA-interim dataset for this analysis based on the current rationale mentioned in line 85-86. Moreover, this could add a clearer link to the current rationale, but depending on the revisions taken by the authors following the above question to clarify this rationale, this might or might not be preferred (anymore).

Furthermore, the sole use of the ONI index for determining the ENSO states, as described in line 87-89 (section 2.1), might provide a relatively poor representation of ENSO events in the paper, so that the difference in spring transition and STT-PBL presented in section 3.3 might be based on an incomplete definition of the ENSO states. Trenberth and Stepaniak (2001) suggest that at least two indices are required to characterize the variability in ENSO events. They advocate that the ONI index should be accompanied by an (orthogonal) index that represents the zonal gradient in sea-surface temperatures (SST). For this purpose, they have created the Trans-Niño Index (TNI), which represents the difference in normalized SST anomalies between the Niño-1+2 and the Niño-4 regions. However, in the paper by Breeden et al. (2020) all positive, neutral and negative ENSO events are lumped into classes, whereas the study of Trenberth and Stepaniak (2001) is also focused on the variability between different occurrences of positive, neutral and negative ENSO events. Therefore, the cruder representation of ENSO events by Breeden et al. (2020), using only the ONI index, might be justified, so that the results in section 3.3 regarding the effect of ENSO states on the spring transition and STT-PBL would not be significantly affected by this approach. Yet, in order to verify whether this approach is indeed justified, I suggest repeating the analysis for section 3.3, regarding the impact of ENSO on the spring transition and STT-PBL, using more than one index to define the three ENSO groups used in the study (e.g. by including some threshold based on the TNI index presented above). This will provide alternative results for this part of the study than can subsequently be compared to the original for statistically significant differences in timing of the spring transition and monthly mean values of the variables in figure 10 for the three ENSO groups. When significant differences are found in this analysis, it suggests that in fact more indices are required to capture the variability in ENSO events and the effect of that on the spring transition and STT-PBL than just the ONI index, which indicates that this reviewed approach is to be preferred over the original based on the findings of Trenberth and Stepaniak (2001).

Minor comments on the paper:

The role of ozone in this paper is somewhat unclear I find. In my regard it constitutes the context of the study and provides potential for further research, but is not part of the study itself. Yet, it is quite broadly mentioned in the methods and conclusions. I would advise to restrict the role of ozone in this paper to the context in the introduction,

Interactive
comment

further research opportunities in the conclusion and perhaps the background for some of the methods.

The resolution of the zonal and meridional wind on pressure levels (2.5 X 2.5°) is larger than any of the components of the JRA-55 dataset that is used for the calculation of these wind variables (Kobayashi et al., 2015). This seems odd to me. I would advise to explain the reasoning behind the resolution of these variables in the data description in section 2.1.

The use of a fixed amount of mass transport for each trajectory in the calculation of the STT-PBL seems a very simplifying assumption that might potentially cause a lot of variation in STT-PBL to be lost without reading the accompanying reference. I would advise to include a short explanation of the background of this method, especially regarding the fact that the variation in STT-PBL is represented by the number of trajectories rather than the mass of them, after you introduced it in section 2.1.

The significance of the results is currently only assessed visually by means of the 95-percent confidence intervals that result from the significance test described in section 2.3. I would suggest including some form of quantitative assessment of this significance in the paper in the form of, for example, a statistical t-test.

Figure 1a seems random and possibly redundant. It only shows the EOF1 pattern for a PC1 larger than $1\sigma$ and not for the other PC1 states and shows a very similar pattern to what is more extensively shown in figure 2. Therefore, I would suggest removing this figure.

I found figure 7 quite time-consuming to grasp fully. This is mainly the result of the layout of the legend I think. I would suggest mentioning the variables of interest before SST in the legend description instead of 'STT/variable' and potentially even to place the description next to the corresponding lines when the available space allows this.

References:

[Figure]

Breeden, M. L., Butler, A. H., Albers, J. R., Sprenger, M., and O'Neil Langford, A. (2020). The Spring Transition of the North Pacific Jet and its Relation to Deep Stratosphere-to-Troposphere Mass Transport over Western North America, Atmospheric Chemistry and Physics Discussions, https://doi.org/10.5194/acp-2020-604, in review

Holton, J. R., Haynes, P. H., McIntyre, M. E., Douglass, A. R., Rood, R. B., and Pfister, L. (1995). Stratosphere–troposphere exchange, Reviews of Geophysics, 33( 4), 403– 439, doi:10.1029/95RG02097

Kobayashi, S., Ota, Y., Harada, Y., Ebita, A., Moriya, M., Onoda, H., Onogi, K., Kamahori, H., Kobayashi, C., Endo, H., Miyaoka, K., Takahashi, K. (2015). The JRA-55 reanalysis: General specifications and basic characteristics. Journal of the Meteorological Society of Japan. Ser. II, 93(1), 5–48, https://doi.org/10.2151/jmsj.2015-001

Trenberth, K. E., and D. P. Stepaniak. (2001). Indices of El Niño Evolution, Journal of Climate, 14, 1697–1701, https://doi.org/10.1175/1520-0442(2001)014<1697:LIOENO>2.0.CO;2.

Seidel, D. J., Zhang, Y., Beljaars, A., Golaz, J.C., Jacobson, A. R., and Medeiros, B. (2012). Climatology of the planetary boundary layer over the continental United States and Europe, Journal of Geophysical Research, 117, D17106, doi:10.1029/2012JD018143

Škerlak, B., Sprenger, M., and Wernli, H. (2014). A global climatology of stratosphere–troposphere exchange using the ERA-Interim data set from 1979 to 2011, Atmospheric Chemistry and Physics, 14(2), 913–937

Troen, I. and Mahrt, L. (1986). A simple model of the atmospheric boundary layer; sensitivity to surface evaporation, Boundary-layer Meteorology, 37, 129–148

---

## Author Comment (AC1) · 10 Dec 2020

Author response to RC#1

This is one of the most polished papers I have reviewed in several years! The investigation is comprehensive, well motivated, and the text and figures are of high quality. It is an excellent contribution to the field and was a pleasure to read. Many thanks to

the authors for the careful and considerable work put into this manuscript! I have only a handful of comments/suggestions outlined below.

R: We appreciate your attention to our manuscript and are very pleased you found the results of interest!

Lines 41-44: It seems that an annual cycle of tropopause altitude would also be relevant here. Perhaps it frequently reaches a minimum in late winter/early spring.

R: We had not though of that, thank you for raising this point. The tropopause is certainly lowest in the extratropics during winter and into early spring (Seidel and Randel 2006), while the maximum in STT-PBL is confined to spring, so it is difficult to ascertain how much the lower tropopause itself aids in STT-PBL. We suspect forcing for regionalized areas of descent are the dominant process responsible for the STT-PBL differences between winter and spring, as stated in Skerlak et al. 2014.

Line 82: Why is 2.5âŮę resolution data used instead of finer data that is available? It seems reasonable to me since JRA-55 is used mainly for context, but a simple justification along those lines would be nice to include.

R: Thank you for raising this point, since we were interested in large-scale zonal wind variability, using 2.5X2.5 degree resolution seemed sufficient, further supported by the fact that the ERA-Interim results at higher resolution yielded nearly identical results, considering PC1 of 200-hPa zonal wind. We included this point in the text (lines 84-85).

Lines 116-127: The mention of "dynamic tropopause" here is sudden. I would recommend simply specifying the range of PV thresholds used in these studies. Otherwise, the discussion could become confusing and is a bit nuanced as is.

R: Thank you for raising this point, we have modified the text to clearly link the 2-PVU boundary to the dynamic tropopause definition (lines 133-134).

Line 180: I recommend citing Rossby wave breaking studies supporting this statement.

R: Thank you for this suggestion, we agree and have included references (lines 205-206).

Line 307: I would recommend also pointing out that the ENSO-jet relationship is consistent with Rossby wave breaking activity given the occurrence of the jet breaks during La NinÌČa.

R: We have added a sentence highlighting this connection (lines 332-333).

Line 339-347: Excellent points here! :)

R: Thank you, we felt it very important to distinguish mass and ozone changes associated with ENSO, given the added complexity when ozone transport is considered.

Please also note the supplement to this comment:
https://acp.copernicus.org/preprints/acp-2020-604/acp-2020-604-AC1-supplement.pdf

---

## Author Comment (AC2) · 10 Dec 2020

RC#2

This study presents the results of a thorough statistical analysis of springtime stratosphere-to-troposphere transport to the boundary layer (STT-PBL) over the western USA, in the context of the jet structure transition between the winter and summer regimes. The authors use wind fields from the ERA-Interim and JRA-55 reanalyses to identify the dominant Pacific wind patterns at the jet level by the way of EOF analysis and employ previously developed methods to calculate STT-PBL and associated diagnostics. They demonstrate that the intensity of springtime STT-PBL is a function of the timing of the jet transitions, with early transitions leading to more intense STT-PBL driven by deeper and more frequent tropopause folds. Furthermore, they show that the transition timing is correlated with ENSO, and carefully investigate the mechanisms involved, by performing a simultaneous analysis of Rossby wave propagation diagnostics, tropopause folds diagnostics, and PBL depth distributions. The paper is really well written, logically constructed and easy to follow, which is no small feat given the complexity of the subject. It also does a great job referencing the sizable body of literature on the topic. I think the analysis and conclusions are very solid and the paper meets all the ACP criteria for publication. Despite my best efforts (it's my job after all;)) I couldn't find almost any issues with the analysis or presentation. It's a great and important paper and I enthusiastically recommend it for publication almost as is. I do have a few very minor suggestions for edits. The most important one concerns the methods section that, I think, would benefit from adding some more details (even if those details can be found elsewhere) that would make the paper more self-contained.

R: Thank you for the feedback and the attention to our manuscript, we have incorporated your suggestions into our revisions, outlined below.

Minor and technical comments

L82-86. What's the vertical resolution of each reanalysis and why is it sufficient for the present purposes, particularly for driving the trajectory model?

R: For JRA-55 reanalysis we only consider zonal wind on a single pressure level, with a focus on large-scale patterns of variability that do not require high vertical resolution to be determined. The ERA-Interim data used to track folds and STT-PBL is on the 60

original hybrid model levels which extend from the surface to 0.1 hPa, vertical resolution suitable enough to identify tropopause folds (Skerlak et al. 2015). We have added this information to the text (lines 88-91).

LL85-89. Can you comment on how the changing observing system (pre-satellite to satellite to more satellites) in JRA-55 impacts the results?

R: Thank you for raising this point; the JRA-55 documentation shows that temperature values across the varying observational record better match observations compared to JRA-25 in the troposphere and lower stratosphere (Kobayashi et al. 2015), suggesting variations across the changing record are minimal. Also, time series of PC1 shows no discernible transition at the start of the satellite record (figure below), suggesting that the seasonal cycle of the jet captured here is not sensitive to any potential changes in the data associated with changes in the observational record. We have included some of this information in the text (lines 88-89).

L91-93. Can you expand this paragraph a bit? This is one of the main tools used here, and I found it hard to get even a general idea of what's being done there without reading Skerlak et al. 2014. For example, how are the trajectories calculated? Perhaps a simple diagram in the supplementary material?

R: Thank you for raising this point. We have expanded the description of how STT-PBL was determined so that it is (hopefully) easier to follow without having to read earlier papers (lines 105-110).

LL 116-118. Is it possible that this is resolution dependent? Can you comment on that?

R: I believe the question is whether the correspondence between the 2-PVU surface and terminus of the fold is a function of resolution. I do not think, even with perfect observations, that there would be an exact correspondence between the boundary of the fold and the 2-PVU surface. Folding really represents a filamentation of the lower tropopause boundary, involving distortion of lower but still near-tropopause PV values

ranging from 1-4 PVU (Skerlak et al. 2015), which we consider as the main reason that slightly lower PV values better correspond to the terminus of the fold. Studies using different reanalysis at different resolutions (e. g., Breeden and Martin 2018; Albers et al. 2018) reflect similar fold structures, also suggesting that resolution does not affect the correspondence between the fold boundary and PV.

LL207-210. I like the idea of stating the main results in the first concise paragraph. However, at the initial pass, it wasn't clear to me if the second and third sentences are the results or something that we already know. How about something like "The main findings are:. . ." after the first sentence?

R: Thank you for this suggestion, we have modified the text to confirm these results are the main results of this section (line 231).

L220. I'm guessing the bimodality in 5b arises from an oscillation between the two phases during the transition period. It's pretty neat. Can you add a one/two-sentence comment on that?

R: This is an interesting idea, but the bimodal distribution is observed during the positive jet phase, not the transitional phase. Still I think your interpretation is correct – indeed, the leading mode of jet variability during winter looks like the transition (e. g., Athanasiadis et al. 2010), so I think the bimodality is related to wintertime variability, including variability that happens to look like the transition. We have added a comment about this in the text (line 246).

L275. Variability reflected in Fig. 3, right? If so, can you reference Fig. 3 explicitly?

R: That is correct, we have included a reference to Figure 3 (line 301).

LL344-345. I'm struggling to understand this sentence. Please rephrase; it looks like an important point is being made there.

R: Thank you, yes this is an important point in distinguishing mass versus ozone transport when ENSO is involved, as ENSO affects both fold frequency (as we show), as

well as the lower stratospheric ozone reservoir (Albers et al. 2018), thereby affecting how much ozone is transported within each fold (which we do not consider in this study). We have clarified the text (lines 372-373).

Fig. 3 caption. The dashed lines are +/-1 sigma, right?

R: The black dashed lines in Figure 3 show the +/- 0.5 sigma values, since the +0.5 sigma value was used to track the spring transition, and +/- 0.5 sigma range defines the 'spring' jet phase (Fig. 2). We have added this description to the caption.

Please also note the supplement to this comment:
https://acp.copernicus.org/preprints/acp-2020-604/acp-2020-604-AC2-supplement.pdf

―――――――――――――――――

[Figure]

[Figure]

[Figure]

**Fig. 1.** EOF1 pattern and PC1 using JRA-55 reanalysis, 1958-2017.

---

## Author Comment (AC3) · 10 Dec 2020

Sebastiaan Heins SC#1 sebastiaan.heins@wur.nl This review was prepared as part of graduate program course work at Wageningen University, and has been produced under supervision of Prof Wouter Peters. The re- view has been posted because of its good quality, and likely usefulness to the authors and editor. This review was not solicited by

the journal. The paper by Breeden et al. (2020) explores the relationship between the spring transition of the north Pacific jet and stratosphere-to-troposphere mass transport to the planetary boundary layer (STT-PBL) over western North-America. Additionally, the spring transition is linked to the state of the El-NinÌČo Southern Oscillation (ENSO). Analyses are based on the JRA-55 and ERA-interim reanalysis datasets. Interannual variability in the spring peak in STT-PBL found in previous studies is shown to relate to the timing of the spring transition, with larger values of STT-PBL for earlier transitions. Finally, ENSO is found to modulate this timing, with earlier transitions being more prevalent for La-NinÌČa conditions and vice versa. This study adds to previous research by providing underlying mechanisms behind the (variability in the) spring peak in STT-PBL over western North-America that was found before. These results suggest that STT-PBL strength can be predicted based on knowledge of the north Pacific jet and ENSO state in the preceding months. This is relevant for, for instance, air quality prediction at the surface as STT-PBL can function as a natural source of ozone in the PBL. In general, the paper is well-written; the structure and headers of the paper are clear and help to understand the research, the figures diversely visualize the results in both maps and time series, the physical mechanisms discussed in the paper are consistently ex- plained well and create a logic story and the results are frequently discussed in light of previous research. Moreover, I think the paper fits nicely in the scope of this journal. Despite the relatively local study area, the implications of this study are thought to be generally applicable in atmospheric sciences and can therefore be extended to other locations around the world that show similar dynamical patterns. Nevertheless, a sig- nificant revision of this paper is required before it can be accepted for publication in my opinion. Most importantly, in the calculation method of STT-PBL described in line 91-96 (sec- tion 2.1) of the paper I miss an assessment of the un- certainty in the calculation that is associated with the sensitivity to parameter choices. Previous studies have shown that this sensitivity might not be insignificant, so that the specific parameter choices could substantially affect the outcome of the results in sec- tion 3.2 and 3.3 of this study. Firstly, Holton et al. (1995) show that the 380 K potential

temperature surface used in the paper by Breeden et al. (2020) as (one of) the defini-tion(s) of the tropopause coincides relatively well with the tropopause in the tropics, but that this value drops for higher latitudes to approximately the 340/350 K potential tem-perature surface in the region that is considered in the paper (western North-America). Secondly, Skerlak et al. (2014) showed that their STT(-PBL) calculation is highly sen-sitive to the choice of minimal residence time of the air in the stratosphere/troposphere before/after a cross- ing of the tropopause. They find that this sensitivity can be very well approximated by a power law with an exponent of -0.5, which means that the STT estimates for a residence time of 24 and 48 hours respectively deviate by as much as 30 percent. The choice is made by Skerlak et al. (2014) to use a constant value of 48 hours for this parameter. This is validated based on the fact that, although the calcu-lated values of STT-PBL are highly sensitive to the parameter value, the geographical distribution of STT-PBL in which the authors of that study are interested is not. How-ever, in the study of Breeden et al. (2020), this validation does not hold anymore as the STT-PBL calculation is applied to the specific region of western North-America instead of it be- ing used to assess the geographical distribution. Thirdly, the STT-PBL calcu-lation also depends on the accuracy of the PBL height forecast, as this influences the number of trajectories that reach the PBL from the stratosphere. To determine the PBL height the critical Richardson number value of 0.25 is used as the criterion for the PBL top (i.e. the transition from turbulent to laminar flow at the top of the PBL). Troen and Mahrt (1986) indicate that this critical value used for the Richardson number does not have a large influence on the PBL height estimation in unstable conditions, but that it does induce variability in PBL height for neutral conditions. Furthermore, according to Seidel et al. (2012) PBL height is especially uncertain over areas with high elevation, which is the case for parts of the study area of the paper by Breeden et al. (2020) due to the presence of the Rocky Mountains. Altogether, I strongly advise including a sensitivity analysis of the results of section 3.2 and 3.3 to the choice of the parameter values used in the calculation of STT-PBL in order to assess the robustness of the cur-rent conclusions. I suggest this sensitivity analysis to include (based on the above dis-
cussion) the potential temperature surface that is taken to represent the tropopause, the minimum residence time of the trajectories that contribute to STT-PBL and both the forecast uncertainty and the forecast value of the PBL height (as a function of the critical Richardson number chosen to represent the top of the PBL).

R: We appreciate these considerations regarding the STT-PBL variable, and it is certain from the original paper, Skerlak et al. 2014, that the parameter choices you outline here affect the STT-PBL variable to some degree. However, it is far from the scope of this paper to perform such sensitivity analyses, which, to some extent, have already been conducted in the study that introduces the dataset. Indeed, using the PBL instead of a fixed pressure level to define deep STT was deliberate and deemed an overall improvement for truly defining deep STT; improving upon the PBL definition used for this definition of transport to see how the results change could be a next step of study. However, our focus is on considering the relative changes of STT-PBL during different jet phases, which we consider unlikely to change even if the parameters mentioned above were modified, since the changes would be applied to the variable in the same way for all three jet phases. If our results were not so easily corroborated with the eddy characteristics and fold characteristics that are not tied to all of these parameter choices, we would perhaps be more concerned about how the STT-PBL variable definition affects our results. However, we consider the strong correlation between STT-PBL and tropopause fold frequency (Fig. 7) particularly encouraging regarding the success of the STT-PBL variable in capturing the transport associated with the correct intended dynamical process. We note that it was not our intent to conduct a sensitivity analysis of a previously-published and readily available product. Since the reviewer is not explicitly linking any of our key conclusions to the STT-PBL parameter choices, we consider it acceptable to use the dataset as published and defer additional sensitivity analysis to future work.

Additionally, the rationale of using of Japanese Reanalysis-55 dataset does not become clear to me from the paper. In line 85-86 (section 2.1) the authors mention that

this dataset is used because of its relatively long record of ENSO events. Yet, the JRA-55 dataset is only used for assessing the characteristics of the spring transition in section 3.1, as is stated in line 352, without considering any influence of ENSO. According to line 86-87 this is because the transport and tropopause fold diagnostics are derived from the ERA-interim reanalysis instead of the JRA-55 data and, therefore, the former is to be used for the analysis of the relationship between ENSO and the spring transition and STT-PBL in order to be consistent in the data used. Therefore, I would like to ask the authors what the exact benefit of using the JRA-55 dataset is and to incorporate the explanation of this in the description of the data in section 2.1. Additionally, it seems to me that the JRA-55 dataset can in fact be used in the analysis of the spring transition for the different ENSO states in section 3.3 (figure 8) as this analysis does not concern any mass transport or tropopause fold characteristics yet and table 1 shows that data on the ENSO states during the spring transition is available for the JRA-55 dataset. Therefore, I would suggest using the JRA-55 dataset instead of the ERA-interim dataset for this analysis based on the current rationale mentioned in line 85-86. Moreover, this could add a clearer link to the current rationale, but depending on the revisions taken by the authors following the above question to clarify this rationale, this might or might not be preferred (anymore).

R: Thank you for raising this point, Figure 8 does indeed show the PC1 response to ENSO for the JRA-55 reanalysis, not ERA-Interim, as you suggested. However, this was not explicitly indicated in the text, which we have now fixed.

Furthermore, the sole use of the ONI index for determining the ENSO states, as described in line 87-89 (section 2.1), might provide a relatively poor representation of ENSO events in the paper, so that the difference in spring transition and STT-PBL presented in section 3.3 might be based on an incomplete definition of the ENSO states. Trenberth and Stepaniak (2001) suggest that at least two indices are required to characterize the variability in ENSO events. They advocate that the ONI index should be accompanied by an (orthogonal) index that represents the zonal gradient in sea-surface

temperatures (SST). For this purpose, they have created the Trans-NinÌČo Index (TNI), which represents the difference in normalized SST anomalies between the NinÌČo-1+2 and the NinÌČo-4 regions. However, in the paper by Breeden et al. (2020) all positive, neutral and negative ENSO events are lumped into classes, whereas the study of Trenberth and Stepaniak (2001) is also focused on the variability between different occurrences of positive, neutral and negative ENSO events. Therefore, the cruder representation of ENSO events by Breeden et al. (2020), using only the ONI index, might be justified, so that the results in section 3.3 regarding the effect of ENSO states on the spring transition and STT-PBL would not be significantly affected by this approach. Yet, in order to verify whether this approach is indeed justified, I suggest repeating the analysis for section 3.3, regarding the impact of ENSO on the spring transition and STT-PBL, using more than one index to define the three ENSO groups used in the study (e.g. by including some threshold based on the TNI index presented above). This will provide alternative results for this part of the study than can subsequently be compared to the original for statistically significant differences in timing of the spring transition and monthly mean values of the variables in figure 10 for the three ENSO groups. When significant differences are found in this analysis, it suggests that in fact more indices are required to capture the variability in ENSO events and the effect of that on the spring transition and STT-PBL than just the ONI index, which indicates that this reviewed approach is to be preferred over the original based on the findings of Trenberth and Stepaniak (2001).

R: We appreciate this consideration, and if the zonal wind differences and STT-PBL differences grouped by ENSO phase were inconsistent with past studies examining the impact of ENSO on the jet (Renwick and Wallace 1996; Shapiro 2001; Breeden et al. 2020) and deep STT (Lin et al. 2015), we would perhaps be concerned by our ENSO definition not grouping the right events. Since the ONI definition is the operational definition used by NOAA, we consider it a reasonable index to use in this study. It would certainly be an interesting avenue of future work given the ongoing research exploring ENSO diversity as reflected by the TNI, and we have included this

point in the conclusions.

Minor comments on the paper:

The role of ozone in this paper is somewhat unclear I find. In my regard it constitutes the context of the study and provides potential for further research, but is not part of the study itself. Yet, it is quite broadly mentioned in the methods and conclusions. I would advise to restrict the role of ozone in this paper to the context in the introduction, further research opportunities in the conclusion and perhaps the background for some of the methods.

R: We agree, and indeed the word ozone does not appear in our results section, but is only used as motivation for the study, a brief point in the methods, and in the discussion.

The resolution of the zonal and meridional wind on pressure levels (2.5 X 2.5âŮę) is larger than any of the components of the JRA-55 dataset that is used for the calculation of these wind variables (Kobayashi et al., 2015). This seems odd to me. I would advise to explain the reasoning behind the resolution of these variables in the data description in section 2.1.

R: Thank you, we have provided this reasoning in the methods per the request of a designated reviewer. We only use EOF1 of 200-hPa zonal wind from this dataset, which would not be sensitive to such differences in resolution.

The use of a fixed amount of mass transport for each trajectory in the calculation of the STT-PBL seems a very simplifying assumption that might potentially cause a lot of variation in STT-PBL to be lost without reading the accompanying reference. I would advise to include a short explanation of the background of this method, especially regarding the fact that the variation in STT-PBL is represented by the number of trajectories rather than the mass of them, after you introduced it in section 2.1.

R: The amount of mass per trajectory actually varies based on the originating gridpoint, it is not fixed for the entire domain but for each gridpoint. The numbers provided in the

text are examples for a trajectory originating in the extratropics (Skerlak et al. 2014). This text has been clarified.

The significance of the results is currently only assessed visually by means of the 95-percent confidence intervals that result from the significance test described in section 2.3. I would suggest including some form of quantitative assessment of this significance in the paper in the form of, for example, a statistical t-test.

R: STT-PBL is not normally distributed (Fig. 5a), and for some groups (ie, La Nina, May days) there are quite small samples once autocorrelation is factored in, degrading the robustness of a t-test.

Figure 1a seems random and possibly redundant. It only shows the EOF1 pattern for a PC1 larger than $1\sigma$ and not for the other PC1 states and shows a very similar pattern to what is more extensively shown in figure 2. Therefore, I would suggest removing this figure.

R: The pattern of the seasonal cycle of u200 has never been presented in the literature to our knowledge, so we consider it important to be shown here.

I found figure 7 quite time-consuming to grasp fully. This is mainly the result of the layout of the legend I think. I would suggest mentioning the variables of interest before SST in the legend description instead of 'STT/variable' and potentially even to place the description next to the corresponding lines when the available space allows this.

References: Breeden, M. L., Butler, A. H., Albers, J. R., Sprenger, M., and O'Neil Langford, A. (2020). The Spring Transition of the North Pacific Jet and its Relation to Deep Stratosphere-to-Troposphere Mass Transport over Western North America, Atmo- spheric Chemistry and Physics Discussions, https://doi.org/10.5194/acp-2020-604, in review Holton, J. R., Haynes, P. H., McIntyre, M. E., Douglass, A. R., Rood, R. B., and Pfister, L. (1995). StratosphereaÌĆA ÌĘRËĞtroposphere exchange, Reviews of Geophysics, 33( 4), 403– 439, doi:10.1029/95RG02097 Kobayashi, S., Ota,

[Figure]

Y., Harada, Y., Ebita, A., Moriya, M., Onoda, H., Onogi, K., Kama- hori, H., Kobayashi, C., Endo, H., Miyaoka, K., Takahashi, K. (2015). The JRA-55 reanalysis: General specifications and basic characteristics. Journal of the Meteoro- logical Society of Japan. Ser. II, 93(1), 5–48, https://doi.org/10.2151/jmsj.2015-001 Trenberth, K. E., and D. P. Stepaniak. (2001). Indices of El NinÌČo Evolution, Journal of Climate, 14, 1697–1701, https://doi.org/10.1175/1520- 0442(2001)014<1697:LIOENO>2.0.CO;2. Seidel, D. J., Zhang, Y., Beljaars, A., Golaz, J.C., Jacobson, A. R., and Medeiros, B. (2012). Climatology of the planetary boundary layer over the continen- tal United States and Europe, Journal of Geophysical Research, 117, D17106, doi:10.1029/2012JD018143 SÌŇkerlak, B., Sprenger, M., and Wernli, H. (2014). A global climatology of stratosphere– troposphere exchange using the ERA-Interim data set from 1979 to 2011, Atmospheric Chemistry and Physics, 14(2), 913–937 Troen, I. and Mahrt, L. (1986). A simple model of the atmospheric boundary layer; sensitivity to surface evaporation, Boundary-layer Meteorology, 37, 129–148